# Impact of DC Electric Field Direction on Sedimentation Behavior of Colloidal Particles in Water

**DOI:** 10.3390/ma18061335

**Published:** 2025-03-18

**Authors:** Hiroshi Kimura

**Affiliations:** Department of Chemistry and Biomolecular Science, Faculty of Engineering, Gifu University, Gifu 501-1193, Japan; kimura.hiroshi.b1@f.gifu-u.ac.jp

**Keywords:** colloidal particles, sedimentation behavior, DC electric field, electric field direction, flocculation, dispersion stability, convective flow, electric double layer

## Abstract

Colloidal particles in water exhibit increased sedimentation velocity under a horizontal DC electric field of several V/mm compared to no field. Hollow particles with a lower density than water show an increased ascent velocity with the horizontal electric field. These phenomena suggest that colloidal particles form flocs due to the electric field, known as the Electrically Induced Rapid Separation (ERS) effect. This study investigates, for the first time, the impact of the DC electric field direction on the ERS effect. The electric field was defined as horizontal when the inclination angle *θ* = 0° and vertical at *θ* = 90°, covering all inclination angles. Results showed that the ERS effect increased for *θ* < ~20–30° in both upward and downward directions. However, beyond this range, the ERS effect decreased or disappeared. At larger *θ* values, convection was observed, significantly improving colloidal particle dispersion stability. Additionally, negatively charged particles were observed to be “repelled” near the negative electrode. This study offers new insights into controlling particle dispersion stability using electric fields and suggests potential applications in colloid and material science.

## 1. Introduction

The dispersion state of colloidal particles in aqueous dispersions has long been of interest from both academic and engineering application perspectives. Smaller colloidal particles tend to achieve a better dispersion state due to more vigorous Brownian motion in water. When considering the dispersion state of sedimenting colloidal particles, the Péclet number *Pe* is an effective parameter for comparing the sedimentation velocity of colloidal particles and the degree of diffusion [1,2]. In this study, the sedimentation behavior of poly(methyl methacrylate) (PMMA) particles (diameter: 5.3 μm) in water was investigated. The PMMA particles used are highly monodisperse and spherical, contributing to the simplicity of the system. The Péclet number is approximately 100, indicating that while the effect of Brownian motion cannot be entirely ignored, it does not significantly influence the sedimentation behavior of the particles. Additionally, the particle Reynolds number *Re*_p_ is 1.7 × 10^−5^, indicating that the sedimentation velocity follows the Stokes approximation. The rotational diffusion coefficient is approximately 0.01 s^−1^. This implies that the particles may undergo random rotational motion on a timescale of around 100 s. However, since PMMA particles are perfectly spherical, their rotational motion can be considered to have negligible influence on the sedimentation velocity of the particles. Therefore, the PMMA aqueous dispersion used in this study is suitable for investigating particle sedimentation behavior. In colloidal dispersion systems, the dispersion state significantly affects macroscopic properties. It is particularly important to consider that an electric double layer forms around colloidal particles in water, which contributes to maintaining a stable dispersion state. The presence of the electric double layer is a key factor in the electrostatic repulsion between colloidal particles. Even in highly dilute colloidal aqueous dispersions, extensive deionization treatment can lead to the formation of colloidal crystals or amorphous solids, which is a well-known phenomenon. For example, Pieranski et al. [3] used ion-exchange resins to reduce the electrolyte concentration to the order of 10^−6^ mol/L to promote the crystallization of colloidal polystyrene particles (diameter: 1.1 μm) with a particle concentration of 0.12 × 10^10^ particles/cm^3^. As the electrolyte concentration increases, the electric double layer shrinks, and when a certain threshold is exceeded, particle aggregation (coagulation or flocculation) occurs. This phenomenon can be clearly explained, particularly in simple systems, by the DLVO theory [4,5]. The effective electrolyte concentration of the dispersing medium used in this study was 1.0 × 10^−5^ mol/L, resulting in a Debye screening length of 96 nm [6]. Although this Debye length is relatively small compared to the particle size of PMMA, it does not significantly affect the sedimentation velocity. However, as discussed later, significant electrostatic repulsion occurs when particles come into close proximity.

The author has demonstrated that the application of a horizontal DC electric field to colloidal aqueous dispersions induces the rapid sedimentation of particles, a phenomenon referred to as the Electrically Induced Rapid Separation (ERS) effect [7,8,9,10]. When comparing the effect under the same electric field strength, a DC electric field produces a significantly greater ERS effect compared to an AC electric field. The ERS effect has been observed in aqueous dispersions containing particles such as polymethyl methacrylate (PMMA), silica particles, bentonite, and other clay minerals, indicating that the ERS effect could also be expected in dispersions composed of various other materials. It is well known that interfacial electrokinetic phenomena, such as electrophoresis, occur when an external electric field is applied. However, despite the importance of the phenomenon of colloidal particle aggregation and redispersion upon the application and removal of an electric field, very few studies have investigated it. On the other hand, numerous studies have explored colloidal particle behaviors in highly confined spaces, particularly near electrode surfaces. For example, Guelcher et al. [11,12] investigated the intriguing aggregation behavior of colloidal particles, including polystyrene particles (diameter 2.5–10 µm), deposited on horizontal electrode surfaces under a vertical electric field. They explained that aggregation progresses due to electroosmotic flow, driven by interactions between the electric double layer on particle surfaces and the external electric field, resulting in the attraction of colloidal particles. Similarly, Fraden et al. [13] reported that polystyrene particles formed chain-like structures under an alternating electric field (30–150 kHz). These chains remained stable under high-intensity electric fields (1000 V/cm), whereas lower field intensities resulted in dynamic equilibrium conditions, characterized by repeated cycles of formation and disintegration. Furthermore, Wei et al. [14] observed irreversible aggregation of polystyrene particles subjected to an alternating electric field (frequency: 1.5 kHz, voltage: 1.3 V, and electrode spacing: 50 µm). They identified two distinct stages in the aggregation process: reaction-limited aggregation (RLA) and diffusion-limited aggregation (DLA). Additional relevant studies include electrophoretic deposition techniques [15,16] and colloidal crystallization [17,18,19]. More recent investigations also present intriguing findings such as controlling particle assembly using rotating electrodes [20], particle rotation induced by DC electric fields (Quincke rotation) [21], and sedimentation and flotation manipulation through hollow particles [22]. Kim et al. [23] have proposed a new numerical simulation method based on the smooth profile (SP) method to accurately and efficiently analyze electrophoretic phenomena in charged colloidal dispersions. This method discusses the influence of an electric field on the electric double layer and dispersion state of colloidal particles. It explains the deformation and overlap of the electric double layer as well as the electrophoretic mobility of particles in dense dispersions. On the other hand, this paper focuses on the phenomenon of particle flocculation, targeting “sedimenting particles” under different electric field conditions.

When the ERS effect occurs, macroscopic observations show that electrophoresis is hardly observed even under a DC electric field, and the particle group in the cell exhibits rapid sedimentation while maintaining a “horizontal” boundary between the colloid-rich turbid region and the water region. However, very recently, the author has reported for the first time that the ERS effect occurs after horizontal electrophoresis when the volume fraction is below 0.0001 [22]. In previous studies, the sedimentation velocity of PMMA particles and montmorillonite (Mt) in water was found to increase up to several hundred times under an applied electric field strength of 1.0 V/mm DC in the particle volume fraction range of 0.0001–0.001 [7]. In mixed aqueous dispersions of particles with different sizes, large and small particles form co-flocs, significantly increasing the sedimentation velocity. At the same time, it was found that slight vibrational fields inhibit the ERS effect [10]. Furthermore, Mori et al. [24] first revealed that the ERS effect depends on pH and is more pronounced when the particle zeta potential is larger, and the author subsequently confirmed similar phenomena [10]. These results indicate that the presence of an electric double layer plays an important role in the manifestation of the ERS effect. It is highly likely that colloidal particles in water form flocs under an electric field. An interesting result supporting this is the ERS effect observed in mixed aqueous dispersions of hollow particles and PMMA particles. Upon application of a horizontal DC electric field, PMMA particles in water exhibit rapid sedimentation, while hollow particles exhibit rapid ascent [9]. This suggests that both PMMA particles and hollow particles form flocs under the electric field. In these mixed systems, it has been clarified that when the proportion of hollow particles is small, all particles sediment under the electric field, and when the proportion of hollow particles is large, all particles ascend [22]. These results were obtained under the condition that the electric field was applied “horizontally” to the dispersion. It has also been shown by the author that the ERS effect greatly depends on the direction of the electric field. The author investigated changes in the dispersion state of PMMA particle aqueous dispersions by applying a vertical DC electric field using a rectangular cell [8]. When horizontal parallel plate electrodes were installed in the cell with the positive electrode at the bottom and the negative electrode at the top, PMMA particles with negative zeta potential exhibited an increase in sedimentation velocity simply due to the “electrophoretic velocity toward the bottom of the cell”. This suggests that the particles did not form flocs. Conversely, when the electric field was applied in the opposite direction, convection of PMMA particles became prominent, and the dispersion state remained uniform even after several hours. Thus, when using a “vertical” DC electric field, different field directions induce opposite phenomena: either rapid sedimentation of particles or suppression of particle sedimentation. As described above, this study focused on the sedimentation behavior of PMMA particles in water under a DC electric field, not in a confined space but in a freely settling system. Furthermore, for the first time, we investigated how the dispersion state of the particles changes when the direction of the DC electric field is varied from horizontal to vertical.

## 2. Experimental Methods

### 2.1. Sample Preparation

Monodisperse polymethyl methacrylate (PMMA) particles were purchased from Sekisui Plastics Co., Ltd., (Osaka, Japan). The particle diameter was 5.3 ± 0.4 µm, with a density of 1.2 g/cm^3^ and a zeta potential of –40 mV [9]. The PMMA particles were dispersed in ultrapure water (Milli-Q Advantage A10, Millipore Co., Burlington, MA, USA) and were subjected to desalination treatment for more than three months using ion-exchange resin (AG501-X8 (D), Bio-Rad Lab., Inc., Hercules, CA, USA). This stock solution was then diluted with ultrapure water to achieve a uniform particle volume fraction of 0.0005. The PMMA particles in water were observed using a digital microscope (BA81-6T-1080M, Shimadzu RIKA Co., Tokyo, Japan).

### 2.2. Observation of the Temporal Changes in the Dispersion State of PMMA Particles

A pair of parallel stainless steel (SUS304) electrodes was installed inside a transparent cubic cell (10 × 10 × 10 mm) (Figure 1a). The distance between the electrodes was 9.8 mm. After the dispersion was introduced into the measurement cell, a rubber cup was attached. The PMMA particles used in this study gradually settled over time, forming a relatively distinct boundary between the colloid-rich turbid region at the bottom and the transparent water region above. In this study, the sedimentation velocity of particles was calculated from the movement of the boundary in the vertical direction passing through the center of gravity of the measurement cell (indicated by a red dot in Figure 1a). Recent preliminary experiments revealed differences in sedimentation velocities depending on the observation height even without an applied electric field. Therefore, we defined three distinct regions: the upper, middle, and lower regions. Four observation points were determined in the vertical direction: specifically, 3.75 mm above the center of gravity (Observation Point I), 1.25 mm above the center (Observation Point II), 1.25 mm below the center (Observation Point III), and 3.75 mm below the center (Observation Point IV). The boundary typically passes through Observation Points I, II, III, and IV in that order as the PMMA particles settle. The distance between adjacent observation points was 2.50 mm, and the particle sedimentation velocity between each observation point was assumed to be constant. On a horizontal platform, the heights of the observation points (*H*_obs_) were 1.25 mm, 3.75 mm, 6.25 mm, and 8.75 mm from the bottom. The measurement cell was quickly secured using double-sided tape on either a horizontal platform or an inclined platform (Figure 1b). The position of the cell’s center of gravity remained fixed, and only the angle was changed. Therefore, when the cell was tilted, it should be noted that *H*_obs_ = 0 does not correspond to the lowest point of the dispersion, and *H*_obs_ = 10 mm is not the highest point. The tilt angle *θ* of the platform was varied from 0° to 90° in increments of 10°. Low-speed imaging was performed using a fixed-point observation camera (TLC200 Pro, Brinno Inc., Taipei, Taiwan) with an interval of 10 s and a frame rate of 10 FPS (Figure 1c). An electric field of *E* = 0.3 V/mm DC was applied to the dispersion using a function generator (FG110, Yokogawa Test & Measurement Co., Tokyo, Japan). This electric field strength was selected based on previous studies by the authors [7,8], which demonstrated that it effectively suppresses electrolysis over extended periods while still inducing the ERS effect. In this study, the electric field direction was defined as “upward electric field” (Figure 1d, left) and “downward electric field” (Figure 1d, right). In the upward electric field, when *θ* = 90°, the electrode plates were horizontally positioned with the positive electrode at the bottom and the negative electrode at the top. Conversely, in the downward electric field, when *θ* = 90°, the negative electrode was positioned at the bottom, and the positive electrode was at the top. The current value was confirmed using a digital multimeter (CDM–16D, CUSTOM Co., Tokyo, Japan). The recorded videos were processed using editing software (Adobe Premiere Pro, Adobe Inc., San Jose, CA, USA) to calculate the sedimentation velocity. All experiments were conducted at 25 °C.

## 3. Results and Discussion

### 3.1. Sedimentation Behavior of PMMA Particles Under No Electric Field

To confirm the stability of the dispersion state of PMMA particles in water, a DLVO potential energy curve was created [4,5] (Figure 2). The DLVO theory applies to systems containing ions in dispersion media with charged particles. The present study targets a deionized aqueous dispersion, where particles are negatively charged—a typical system described by DLVO theory. The DLVO potential energy curves exhibit complex variations depending on particle size, surface potential (zeta potential), and electrolyte concentration. The procedure for creating this curve follows the author’s previous research [21], including the values for the Hamaker constant and the Debye screening length. The Hamaker constant value of 6.3 × 10^−20^ J was used for PMMA particles. According to the previous study, the Debye screening length was 96 nm. The vertical axis of the potential energy curve represents the total potential energy divided by the thermal energy, indicating how large the interaction energy between particles is compared to thermal motion. It is known that if the potential barrier exceeds 25, aggregation is prevented [25]. The DLVO potential energy curve indicates that there is a sufficiently large electrostatic repulsion between PMMA particles, preventing aggregation and confirming that individual particles settle.

The particle Reynolds number *Re*_p_ is expressed by the following equation [26]:(1)$${Re}_{p}=\frac{\rho_{w}vd}{\eta }$$
where *ρ*_w_ is the density of water, *v* is the particle sedimentation velocity, *d* is the particle diameter, and *η* is the viscosity of water (0.00089 Pa⋅s at 25 °C). The value of *Re*_p_ was 1.7 × 10^−5^, confirming that the Stokes approximation holds for the sedimentation velocity of PMMA particles. The Péclet number *Pe* is given by the following equation [27]:(2)$$Pe=\frac{\pi d^{4}\Delta \rho g}{12k_{B}T}$$
where Δ*ρ* is the density difference between the particles and water, *g* is the acceleration due to gravity, *k*_B_ is the Boltzmann constant, and *T* is the absolute temperature. The Péclet number for the PMMA particles used in this study was 99, suggesting that sedimentation dominates over diffusion, resulting in the formation of a relatively sharp boundary due to particle settling. The above results can be summarized as follows:▪ In the absence of an electric field, interference with surrounding particles can be ignored, and particles settle without forming flocs.▪ The sedimentation velocity of PMMA particles follows the Stokes approximation in a stationary fluid.▪ The descent velocity of the boundary can be considered equal to the sedimentation velocity of the PMMA particles.

Based on the Stokes approximation, the sedimentation velocity *v* of a sphere in a stationary fluid is expressed as follows:(3)$$v=\frac{d^{2}(\rho_{p}–\rho_{w})g}{18\eta }$$
where *ρ*_p_ is the particle density. Consequently, the Stokes value for the sedimentation velocity of PMMA particles in water (*d* = 5.3 µm) is 3.4 µm/s.

### 3.2. Sedimentation Velocity of PMMA Particles Under No Electric Field and Applied Electric Field

First, the dependence of the sedimentation velocity of PMMA particles under no electric field on the inclination angle *θ* was investigated (Figure 3a). The measured values at *θ* = 0 and *θ* = 90° were 2.9 µm/s, which is close to the Stokes value of 3.4 µm/s. When comparing the sedimentation velocity in the upper, middle, and lower regions of the dispersion, the upper region showed the highest velocity, followed by the middle and lower regions. This can be attributed to increased particle interactions in the lower region, as it is known that the stability of particle dispersion improves with an increase in the particle volume fraction.

Examining the sedimentation velocity over the full range of inclination angles (*θ* = 0–90°), a convex upward curve was observed, particularly in the upper region. Even at around *θ*~50°, where the sedimentation velocity was highest in the upper region, the boundary remained almost horizontal, from the start of observation, while descending. It is interesting to note that the particle group near the peak instantly forms a horizontal boundary. Additionally, the particles were found to settle at a velocity approximately 1.5 times greater than the Stokes value. Heitkam et al. [28] reported that convection near the walls could either increase or decrease the sedimentation velocity of particles, suggesting that convection phenomena near the walls may contribute to the observed increase in sedimentation velocity. The height dependence and inclination angle dependence of the sedimentation velocity under no electric field were found to be considerably smaller compared to the changes observed with the application of an electric field. Still images were created from recorded videos and summarized (Figure 4). Representative images for *θ* = 0°, 30°, 50°, 70°, and 90° are shown, while images for other angles are summarized in Appendix A. These still images capture moments when the boundary passes through the four observation points (excluding *t* = 0). While the boundary often remains horizontal, there were instances where it tilted slightly under certain conditions (e.g., Figure 4i,n). Even in the absence of an electric field, turbulence near the walls in the relatively narrow, confined space (1 cm^3^) appeared to have a certain degree of influence on particle sedimentation behavior.

When a positive electric field (*E* = +0.3 V/mm) was applied (Figure 3b), the sedimentation velocity increased significantly compared to no electric field. As in the case without an electric field, the upper region had the highest sedimentation velocity, followed by the middle and lower regions. The sedimentation velocity tended to increase as *θ* increased. This tendency is considered to be influenced by the downward-directed electrophoresis, as discussed later. As *θ* approached 90°, the increase in sedimentation velocity was particularly pronounced in the upper region. Here, still images for the same *θ* conditions as under no electric field are summarized (Figure 5), and images for other angles are compiled in Appendix A. When an electric field was applied, a phenomenon where particles were “swept up” near the boundary was observed, particularly at inclined angles (e.g., Figure 5i,s). This phenomenon resembled swirling snow being swept up by a strong wind. Additionally, attention was paid to the changes in the dispersion state of particles every 60 s when *θ* = 90° (Figure 6a–h). Considering the data in Figure 3b, it should be noted that the sedimentation velocity is particularly high in the upper region for *θ* = 90°, but it was difficult to clearly observe this fact. Therefore, more detailed changes in the boundary height were investigated (Figure 6i). The height of the boundary was found to decrease rapidly in the initial stage and converge to a constant rate of decrease after *t*~200 s. During this time, negatively charged PMMA particles appeared to be strongly “repelled” by electrostatic repulsion, as the horizontal negative electrode was located at the topmost part of the dispersion.

When a negative electric field (*E* = −0.3 V/mm) was applied (Figure 3c), the behavior differed significantly from that under a positive electric field. In all regions (upper, middle, and lower), the sedimentation velocity decreased as *θ* increased. For *θ* < 30°, the sedimentation velocity was greater than under no electric field, but at *θ*~30°, it became almost the same as under no electric field, and for *θ* > 30°, the sedimentation velocity decreased, approaching zero as *θ* increased. In this case, the sedimentation velocity in the lower region was lower compared to higher observation regions for *θ* < 30°. However, for *θ* > 30°, the sedimentation velocities in the upper, middle, and lower regions were almost identical. This observation is summarized in Figure 7, and images for other angles are presented in Appendix A. A phenomenon where particles were “swept up” near the boundary was frequently observed under a negative electric field.

To intuitively capture the differences with and without the application of an electric field, still images at *t* = 500 s were summarized (Figure 8). Here, *θ* = 0°, 30°, 50°, 70°, and 90° were used as examples. Compared with no electric field, it was confirmed that the sedimentation velocity was clearly greater when a positive electric field was applied, even with convection present. Conversely, when a negative electric field was applied, particle sedimentation was clearly suppressed due to convection. Particularly at *θ* = 80° and 90°, a stable dispersion state was maintained up to the maximum observation time (9650 s). The current flowing through the dispersion during the application of an electric field was constant at 53 mA, regardless of the tilt angle or elapsed time.

### 3.3. Relationship Between Electrophoretic Velocity and Sedimentation Velocity

To discuss the effect of electrophoresis on particle sedimentation velocity, the sedimentation velocities under various electric field conditions were compared (Figure 9). In this analysis, data from the middle region were used. In the middle region, the sedimentation velocity under no electric field remained almost constant regardless of the *θ* value (indicated by black circles in the figure). When a positive electric field was applied (Figure 9a), negatively charged PMMA particles experienced an electrically induced vertical downward velocity component *v*_ep,v_ when the cell was inclined.(4)$$v_{ep,v}=\frac{\varepsilon \zeta E}{\eta }\sin\theta$$
where *ε* is the permittivity of water and *ζ* is the zeta potential of the PMMA particles. The gray solid line represents the sum of *v*_ep,v_ and the measured sedimentation velocity under no electric field (2.9 µm/s). If only the forces due to gravity and the vertical component of electrophoresis acted on individual PMMA particles, the calculated values (gray solid line) should match the measured sedimentation velocities. However, in practice, for *θ* < 30°, the measured sedimentation velocity was greater than the calculated values, while for *θ* > 30°, the measured sedimentation velocity was smaller. This result suggests that, in addition to the forces of gravity and electrophoresis acting on individual PMMA particles, there are factors that increase the sedimentation velocity and factors that decrease it. The most straightforward explanation for the increase in sedimentation velocity is the flocculation of sedimenting particles. Flocculation of colloidal particles under an electric field has been proposed by Mori et al. [24]. Additionally, the author observed flocs deposited on a cover glass under an electric field using an optical microscope [9]. Assuming that the floc is spherical and that its sedimentation velocity follows the Stokes approximation, the floc volume corresponds to about four PMMA particles at *θ* = 0° and about two particles at *θ* = 10°. It is thought that particles are more strongly “repelled” by the negative electrode in the upper region, particularly as *θ* approaches 90°, leading to an increase in sedimentation velocity (Figure 3b, data represented by diamond symbols). These results indicate that the agreement between the calculated and measured values at *θ* = 90° in the middle region (Figure 9a) is coincidental. On the other hand, a factor that reduces sedimentation velocity is the presence of convection within the dispersion. Under a negative electric field (*E* = −0.3 V/mm DC, Figure 9b), particles experience an electrically induced vertical force in the opposite direction compared to a positive electric field. For conditions where *θ* < 30°, a similar increase in sedimentation velocity was observed for both positive and negative electric fields. This strongly suggests that the increase in sedimentation velocity occurs through the same mechanism, specifically, the flocculation of particles. For *θ* > 30°, sedimentation velocity increased with *θ* under a positive electric field, clearly indicating that electrophoresis contributed to this increase. However, it is likely that the generation of convection partially counteracted this increase. On the other hand, under a negative electric field, as mentioned earlier, sedimentation velocity continued to decrease toward zero as *θ* increased. Convection under a negative electric field was significantly more intense compared to that under a positive electric field. When a positive electric field was applied, the electrophoretic force acted in the same direction as gravitational sedimentation, whereas under a negative electric field, this force acted in the opposite direction (vertically upward) against particle sedimentation. It is presumed that the balance between these opposing forces resulted in the generation of intense convection. In addition to differences in convection intensity, the “repulsion” phenomenon near the negative electrode also differed between positive and negative electric fields. Under a positive electric field, as mentioned earlier, a clear “repulsion” effect was observed near the negative electrode. However, under a negative electric field, macroscopic observations did not confirm such a “repulsion” effect near the negative electrode. This suggests that the downward gravitational force acting on the particles and the electrostatic repulsion force were balanced. Even for *θ* > 30°, floc formation may still occur. However, even if flocs are momentarily formed, it is likely that convection can easily disrupt them. In this case, floc formation, disruption, and reformation would be continuously repeated. This is because flocs formed under an electric field are extremely weak and, for example, they disappear and redisperse when the electric field is removed [8]. Additionally, even under an electric field, flocs can be disrupted by slight vibrations [10]. In this process, flocs are likely to undergo instantaneous rotational motion due to convection, which may facilitate the dispersion of individual PMMA particles. At the same time, the rotational motion of individual particles itself is expected to act in a direction that suppresses the sedimentation velocity. Therefore, for *θ* > 30°, the stabilization of the dispersion state due to convection is expected to dominate. This study revealed, for the first time, that by controlling the direction of the applied electric field, it is possible to freely manipulate the dispersion stability of colloidal particles in water.

## 4. Conclusions

In this study, the author focused on the sedimentation behavior of settling PMMA particles in water under a DC electric field. For the first time, the author investigated how the dispersion state of the particles changes when the direction of the DC electric field is varied from horizontal to vertical. When a horizontal electric field (*θ* = 0°) was applied, the colloidal particles rapidly sedimented due to the Electrically Induced Rapid Separation (ERS) effect, strongly suggesting floc formation. On the other hand, when a vertical electric field (*θ* = 90°) was applied, the sedimentation of the particles varied significantly depending on the direction of the field. For both cases where the electric field was directed upward with increasing *θ* (defined as a positive field) and downward (defined as a negative field), the ERS effect increased within the range of *θ* < approximately 20–30°. However, for larger *θ* values, the ERS effect decreased or disappeared. It was found that as *θ* increased, convection was induced, leading to a significant improvement in the dispersion stability of colloidal particles. Notably, under a positive electric field at *θ* = 90°, a phenomenon was observed in which negatively charged particles were rapidly “repelled” near the cathode at the initial stage of field application, temporarily increasing the sedimentation rate near the cathode. These results indicate that by appropriately controlling the direction of the electric field, the dispersion stability of colloidal particles can be freely adjusted. Considering the conditions for the ERS effect and the influence of convection caused by the electric field direction, it is suggested that the intentional control of particle sedimentation can be applied to enhance dispersion stability or promote sedimentation. Furthermore, when magnetic particles are present in an aqueous dispersion, the combined application of an electric field and a magnetic field is expected to contribute to the development of new particle control techniques.

## Figures and Tables

**Figure 1 materials-18-01335-f001:**
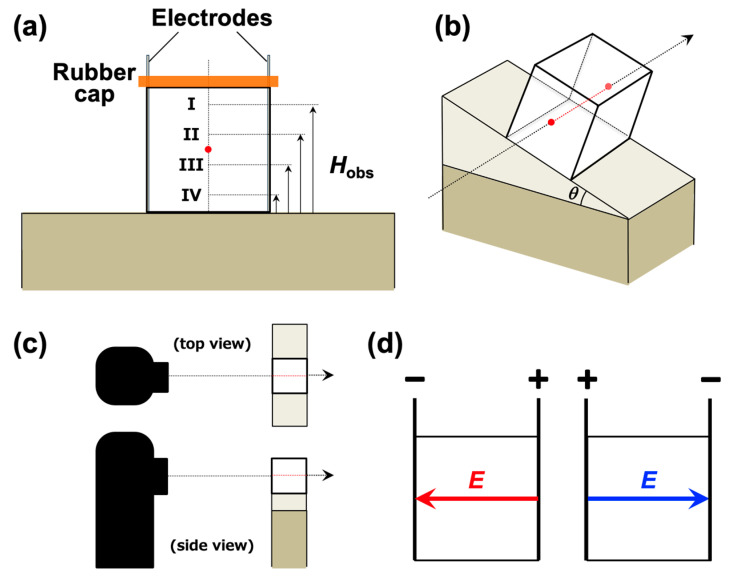
The measurement cell and setup for observing the dispersion state of PMMA particles. (**a**) The schematic diagram of the measurement cell. The cell was tilted using a horizontal line passing through the center of gravity as the rotation axis. Regardless of the inclination angle, the observation points were determined as follows along the vertical line passing through the center of gravity: 3.75 mm above (Observation Point I), 1.25 mm above (Observation Point II), 1.25 mm below (Observation Point III), and 3.75 mm below (Observation Point IV). The observation point heights *H*_obs_ at *θ* = 0° and *θ* = 90° were as follows: *H*_obs_ = 8.75 mm (Observation Point I), 6.25 mm (Observation Point II), 3.75 mm (Observation Point III), and 1.25 mm (Observation Point IV). (**b**,**c**) The inclined platform and observation direction. Observations were made along the horizontal line passing through the center of gravity of the cell. (**d**) The positive electric field (left) and negative electric field (right).

**Figure 2 materials-18-01335-f002:**
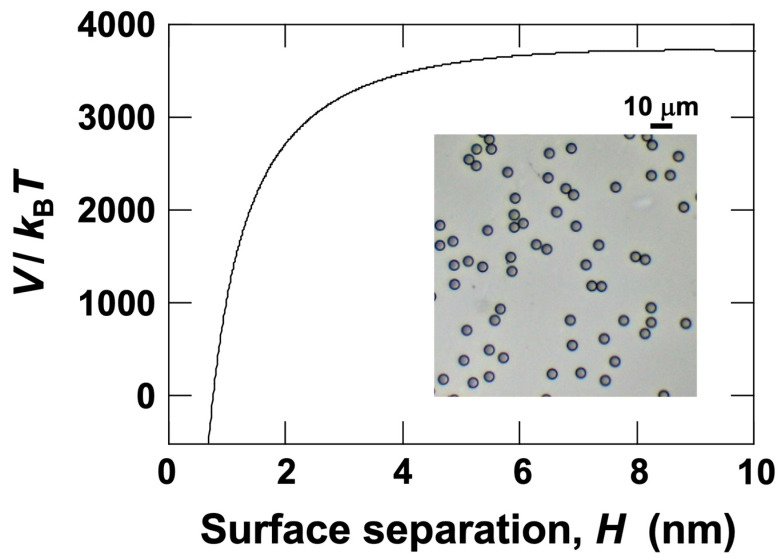
The DLVO potential energy curve for PMMA particles in water. The Hamaker constant *A*_H_ = 6.3 × 10^−20^ J, particle diameter *d* = 5.3 μm, zeta potential *ζ* = −40 mV, Debye screening length *L*_D_ = 96 nm, and absolute temperature *T* = 298 K. The image in the figure shows PMMA particles in water at 25 °C, captured using a digital microscope.

**Figure 3 materials-18-01335-f003:**
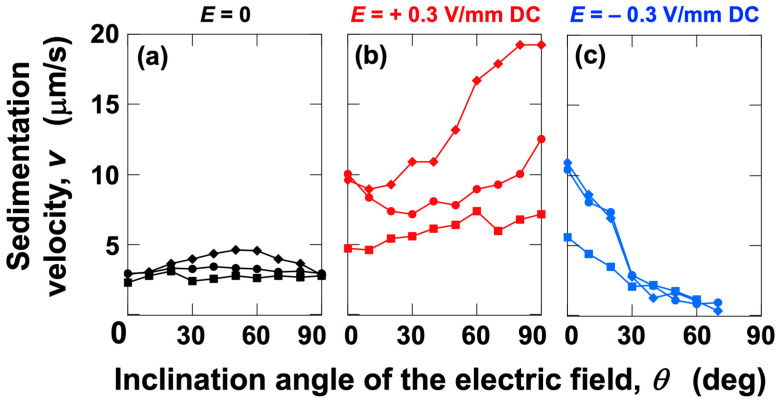
Effect of the electric field direction on the sedimentation velocity of PMMA particles at 25 °C. (**a**) No electric field, (**b**) *E* = +0.3 V/mm DC, and (**c**) *E* = −0.3 V/mm DC. Diamond: upper region of the cell (Observation Points I–II); circle: middle region (Observation Points II–III); and square: lower region (Observation Points III–IV).

**Figure 4 materials-18-01335-f004:**
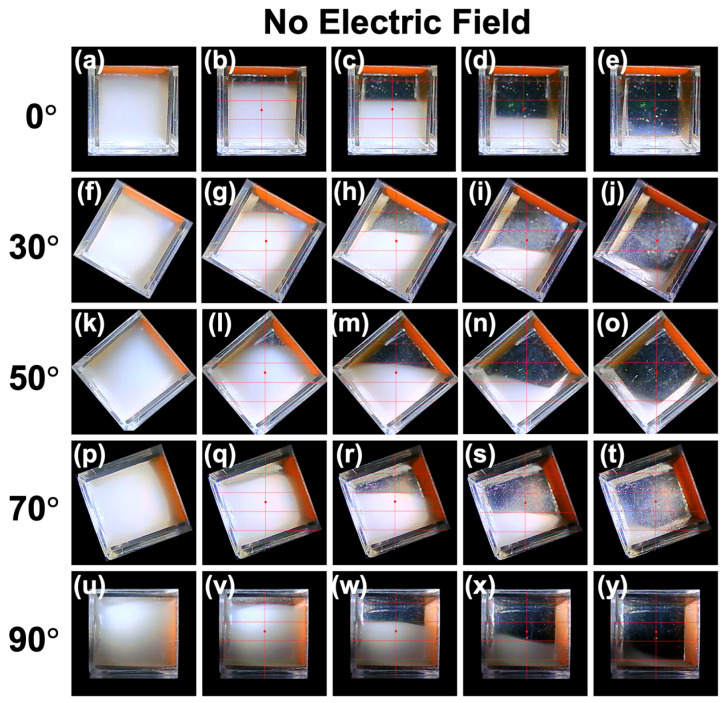
Changes in the dispersion state of PMMA particles at 25 °C under no electric field. (**a**–**e**) Inclination angle *θ* = 0°, (**f**–**j**) *θ* = 30°, (**k**–**o**) *θ* = 50°, (**p**–**t**) *θ* = 70°, and (**u**–**y**) *θ* = 90°. (**a**) Elapsed time *t* = 0, (**b**) 410 s, (**c**) 1280 s, (**d**) 2140 s, (**e**) 3240 s, (**f**) 0, (**g**) 310 s, (**h**) 940 s, (**i**) 1710 s, (**j**) 2770 s, (**k**) 0, (**l**) 270 s, (**m**) 810 s, (**n**) 1560 s, (**o**) 2460 s, (**p**) 0, (**q**) 190 s, (**r**) 820 s, (**s**) 1650 s, (**t**) 2550 s, (**u**) 0, (**v**) 190 s, (**w**) 1080 s, (**x**) 1940 s, and (**y**) 2850 s.

**Figure 5 materials-18-01335-f005:**
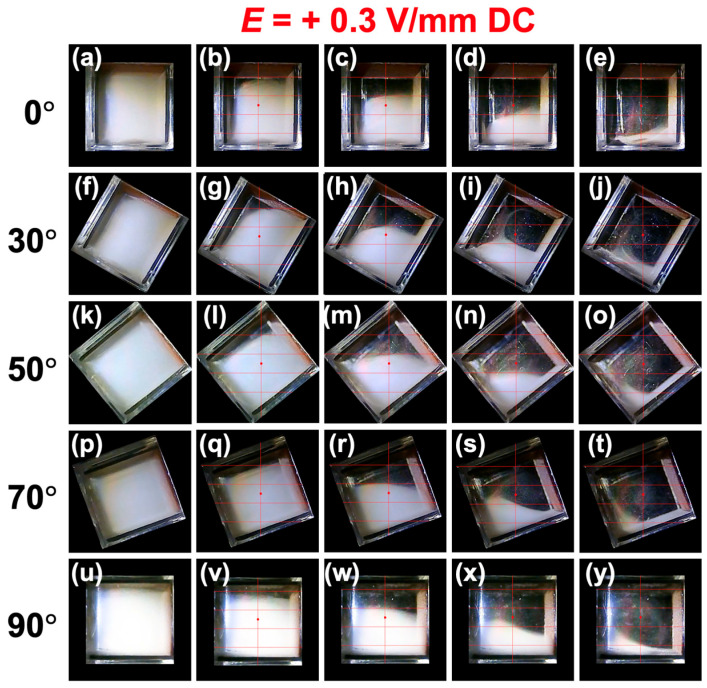
Changes in the dispersion state of PMMA particles at 25 °C under *E* = +0.3 V/mm DC. (**a**–**e**) Inclination angle *θ* = 0°, (**f**–**j**) *θ* = 30°, (**k**–**o**) *θ* = 50°, (**p**–**t**) *θ* = 70°, and (**u**–**y**) *θ* = 90°. (**a**) Elapsed time *t* = 0, (**b**) 180 s, (**c**) 440 s, (**d**) 690 s, (**e**) 1220 s, (**f**) 0, (**g**) 70 s, (**h**) 300 s, (**i**) 650 s, (**j**) 1100 s, (**k**) 0, (**l**) 60 s, (**m**) 250 s, (**n**) 570 s, (**o**) 960 s, (**p**) 0, (**q**) 50 s, (**r**) 190 s, (**s**) 460 s, (**t**) 880 s, (**u**) 0, (**v**) 60 s, (**w**) 190 s, (**x**) 390 s, and (**y**) 740 s.

**Figure 6 materials-18-01335-f006:**
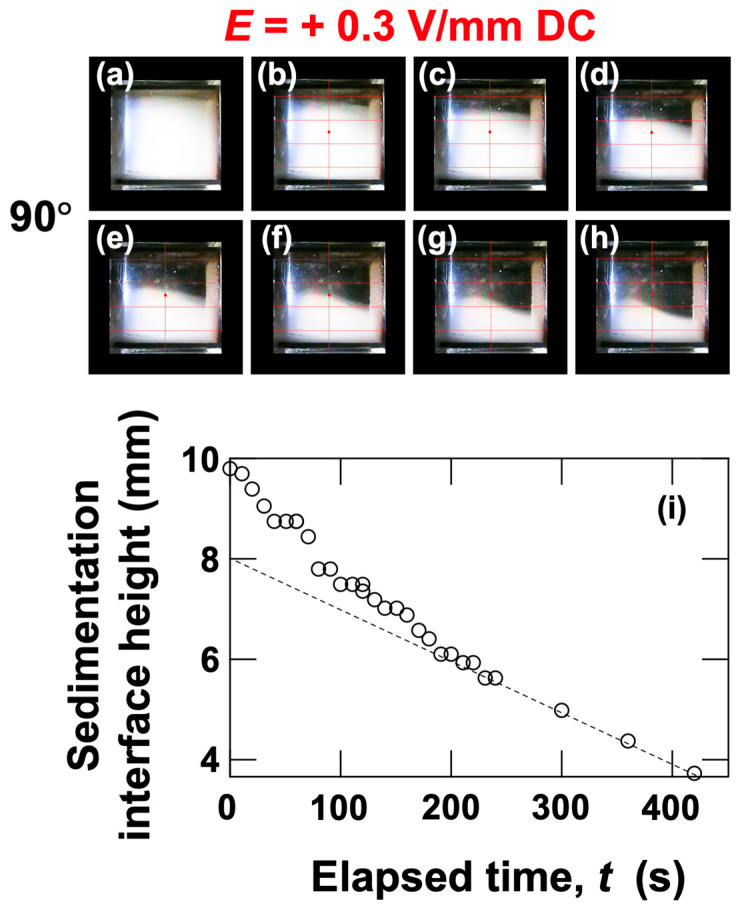
Changes in the dispersion state of PMMA particles at 25 °C under a positive electric field (*E* = +0.3 V/mm DC). Inclination angle *θ* = 90°. (**a**–**h**) Appearance of the dispersion state. (**i**) Changes in the height of the boundary associated with the sedimentation of PMMA particles. (**a**) Elapsed time *t* = 0, (**b**) 60 s, (**c**) 120 s, (**d**) 180 s, (**e**) 240 s, (**f**) 300 s, (**g**) 360 s, and (**h**) 420 s. The dashed line indicates the slope of the change in boundary height after approximately 200 s.

**Figure 7 materials-18-01335-f007:**
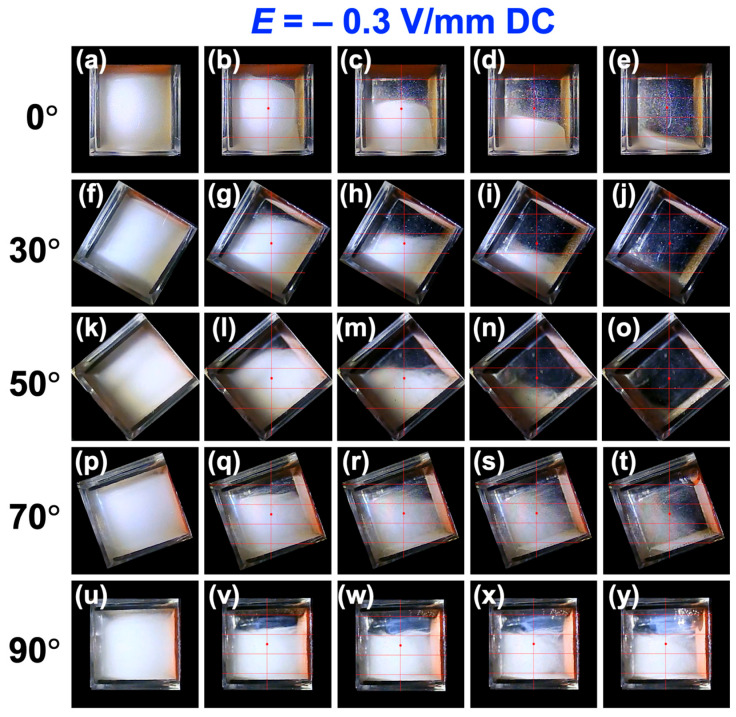
Changes in the dispersion state of PMMA particles at 25 °C under *E* = −0.3 V/mm DC. (**a**–**e**) Inclination angle *θ* = 0°, (**f**–**j**) *θ* = 30°, (**k**–**o**) *θ* = 50°, (**p**–**t**) *θ* = 70°, and (**u**–**y**) *θ* = 90°. (**a**) Elapsed time *t* = 0, (**b**) 100 s, (**c**) 330 s, (**d**) 570 s, (**e**) 1020 s, (**f**) 0, (**g**) 530 s, (**h**) 1430 s, (**i**) 2300 s, (**j**) 3510 s, (**k**) 0, (**l**) 480 s, (**m**) 2020 s, (**n**) 4320 s, (**o**) 5780 s, (**p**) 0, (**q**) 2000 s, (**r**) 4000 s, (**s**) 6000 s, (**t**) 9650 s, (**u**) 0, (**v**) 2000 s, (**w**) 4000 s, (**x**) 6000 s, and (**y**) 9650 s.

**Figure 8 materials-18-01335-f008:**
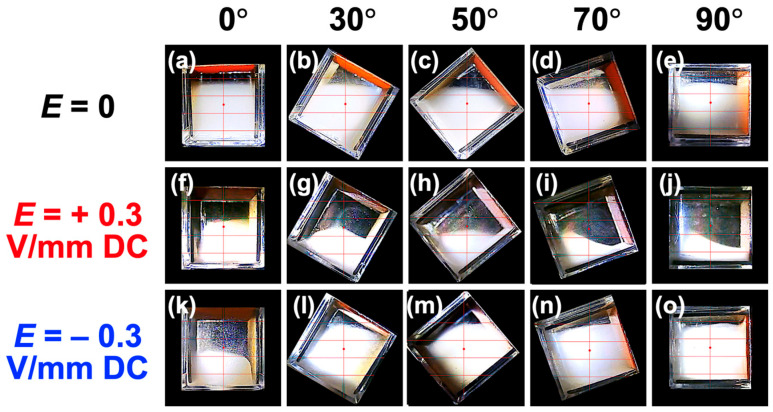
The effect of electric field direction on the dispersion state of PMMA particles at 25 °C. *t* = 500 s. (**a**–**e**) *E* = 0, (**f**–**j**) *E* = +0.3 V/mm DC, and (**k**–**o**) *E* = −0.3 V/mm DC. (**a**,**f**,**k**) Inclination angle *θ* = 0°, (**b**,**g**,**l**) *θ* = 30°, (**c**,**h**,**m**) *θ* = 50°, (**d**,**i**,**n**) *θ* = 70°, and (**e**,**j**,**o**) *θ* = 90°.

**Figure 9 materials-18-01335-f009:**
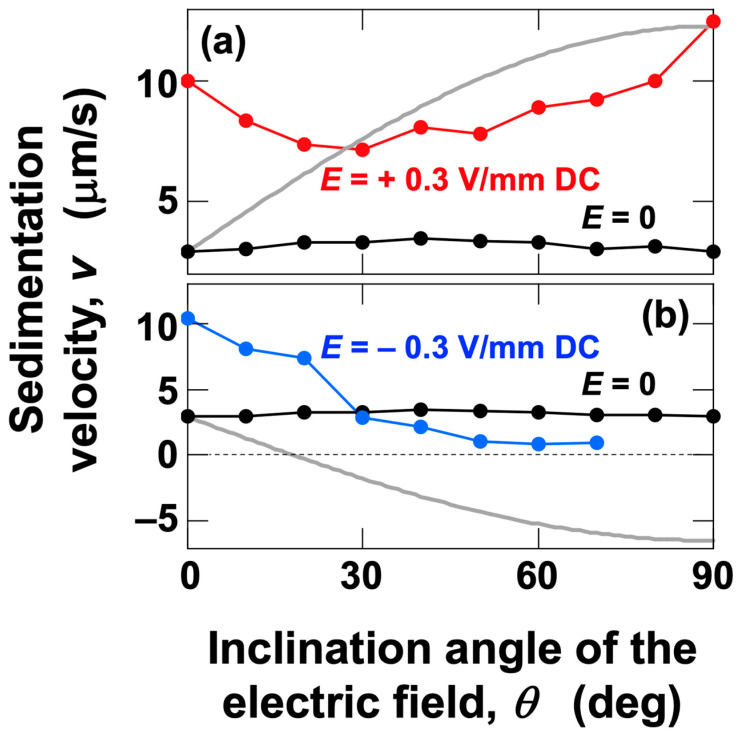
Dependence of the sedimentation velocity of PMMA particles on the direction of the electric field at 25 °C. (**a**) *E* = +0.3 V/mm DC, and (**b**) *E* = −0.3 V/mm DC. The black solid line in the figure represents the sedimentation velocity under no electric field. The gray solid line represents the sum of the vertical component of the electrophoretic velocity and the sedimentation velocity under no electric field (*θ* = 0°, 2.9 µm/s).

## Data Availability

The original contributions presented in this study are included in the article and Appendix A. Further inquiries can be directed to the corresponding author.

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
