# Peer review of "Impact of DC Electric Field Direction on Sedimentation Behavior of Colloidal Particles in Water"

_materials, 2025, doi:10.3390/ma18061335_

Round 1
Reviewer 1 Report
Comments and Suggestions for Authors
The manuscript ID materials-3532428 mainly presents a study about the potential influence of a DC electric field direction on an induced sedimentation behavior exhibited by particular colloidal particles in water. Please see below a list of comments to the authors:
- Please justify the motivation for the selection of poly(methyl methacrylate) (PMMA) particles in this study.
- How was selected the size of the particles in this research?
- Is there a potential agglomeration of the particles during the sedimentation process?
- How was proposed the magnitude of the electric field in the experiment? Please argue.
- Is it considered that the particles are rotating during the sedimentation? If yes, how can be this rotation an influence in the sedimentation?
- Please describe about statistics and reproducibility in the sedimentation process.
- Experimental error bar must be provided.
- Perspectives should be discussed. The authors are invited to discuss about advantages and disadvantages of electric field vs electromagnetic field in the evolution of an induced sedimentation process of particles. You can see for instance: https://doi.org/10.1016/j.ijleo.2021.167738
- The main results should be confronted with updated publication in the topic. You can see for instance: https://doi.org/10.1021/acs.langmuir.1c02581
- It is suggested to split the citations reported in collective form within the text. This is in order to easily visualize the justification and importance of each reference selected for the bibliography of this topic.
A proofreading is suggested
Author Response
Response to Reviewer 1
I sincerely appreciate the detailed and constructive comments provided by the reviewers. I have carefully revised my manuscript to address each point raised. Below, I present my responses to each comment individually.
Comment 1: Justification for Selecting PMMA Particles
Please justify the motivation for the selection of poly(methyl methacrylate) (PMMA) particles in this study.
Response:
I selected poly(methyl methacrylate) (PMMA) particles because they are highly monodisperse and spherical, simplifying the experimental system. This clarification has been included in Lines 33–34 of the revised manuscript.
Comment 2: Criteria for Particle Size Selection
How was the particle size selected in this research?
Response:
I utilized spherical PMMA particles with an approximate diameter of 5 µm and a density of 1.2 g/cm³. These particles yield low particle Reynolds numbers, ensuring sedimentation velocities closely adhere to Stokes' law. This simplifies experimental conditions and effectively highlights the influence of the electric field, as elaborated in Lines 34–39.
Comment 3: Potential Particle Aggregation During Sedimentation
Is there a potential agglomeration of the particles during the sedimentation process?
Response:
Under conditions with no electric field, significant electrostatic repulsion prevents particle aggregation. This point has been clearly discussed in Lines 210–215.
Comment 4: Rationale for Electric Field Strength
How was the magnitude of the electric field determined in the experiment? Please argue.
Response:
The applied electric field intensity (0.3 V/mm DC) was chosen based on previous findings demonstrating its effectiveness in inducing the ERS effect while suppressing electrolysis over extended periods. This rationale is clarified in Lines 174–177.
Comment 5: Consideration of Particle Rotation
Was particle rotation considered during sedimentation? If so, how could this rotation influence the sedimentation process?
Response:
Particle rotation was not explicitly considered. If rotation occurs, formed flocs are likely to disintegrate and disperse individually, a process observed even under minor vibrations (Lines 430–433).
Comment 6: Statistical Treatment and Reproducibility
Please describe the statistical treatment and reproducibility of the sedimentation process.
Response:
Sedimentation velocities were carefully determined from video recordings of interface movements under controlled conditions. In the absence of external disturbances such as leaks, each experiment yielded precise, reproducible data from single video recordings.
Comment 7: Error Bars for Experimental Results
Experimental error bars must be provided.
Response:
Since sedimentation velocities were determined from single video recordings under each condition without replicate measurements, error bars could not be presented. I explicitly acknowledge this limitation.
Comment 8: Discussion of Future Perspectives
Future perspectives should be discussed. The authors are invited to discuss the advantages and disadvantages of electric fields versus electromagnetic fields in the context of induced particle sedimentation. You can see for instance: –
Response:
I reviewed the suggested paper and acknowledge its quality and relevance; however, it does not directly align with the specific objectives of my study, which focuses on understanding sedimentation under controlled directional electric fields. Therefore, I have opted not to cite this paper.
Comment 9: Comparison with Latest Research Findings
The main results should be compared with recent publications on this topic. You can see for instance: –
Response:
I agree with the recommendation. The study on rolling motion (Quincke rotation) under a DC electric field was insightful. This work has been incorporated and cited as reference [21] in Lines 84–86.
Comment 10: Individual Citation of References
It is suggested to split the citations reported in collective form within the text. This is in order to easily visualize the justification and importance of each reference selected for the bibliography of this topic.
Response:
I revised the manuscript by individually addressing the previously grouped citations [11–20]. These have been reorganized in Lines 68–87, including the newly added citation [21], to explicitly clarify the context and significance of each reference.
Thank you again for your insightful comments, which have significantly enhanced the clarity and quality of our manuscript. I hope our responses have adequately addressed your concerns.
Sincerely,
Hiroshi Kimura
Reviewer 2 Report
Comments and Suggestions for Authors
The manuscript investigates the impact of DC electric field orientation on sedimentation behavior of PMMA colloidal particles in water. A pronounced electrostatic repulsion near negative electrodes significantly influenced sedimentation patterns. The research contributes novel insights on how precisely controlling electric field direction can manipulate colloidal dispersion, with implications for advanced material processing technologies.
- The detailed description of the electrode and observation points is thorough, but the rationale behind selecting specific observation heights could be clearer.Briefly justify the chosen observation points to clarify their relevance in analyzing sedimentation behavior.
- The application of DLVO theory is mentioned briefly but lacks detailed contextualization regarding its limitations or assumptions for PMMA particles.
- The mechanism behind particle flocculation induced by electric fields is discussed but not visually demonstrated.
- Incorporate quantitative analysis (e.g., computational fluid dynamics simulations or tracer particles) to directly measure and validate convection patterns.
Author Response
Response to Reviewer 2
I sincerely appreciate your valuable comments and constructive suggestions on my manuscript. I have carefully revised the manuscript according to your insightful remarks. Below, please find my detailed responses to each of your comments.
Comment 1: Clarification of Observation Height Selection
The detailed description of the electrode and observation points is thorough, but the rationale behind selecting specific observation heights could be clearer. Briefly justify the chosen observation points to clarify their relevance in analyzing sedimentation behavior.
Response:
Thank you for pointing out this ambiguity. Previously, observations were conducted at only one specific height based on preliminary studies. However, recent preliminary experiments revealed variations in sedimentation velocity at different observation heights even without an applied electric field. Therefore, I have explicitly described in Lines 154–157 the rationale behind choosing three distinct observation regions: upper, middle, and lower.
Comment 2: Deepening the Discussion on DLVO Theory Application
The application of DLVO theory is mentioned briefly but lacks detailed contextualization regarding its limitations or assumptions for PMMA particles.
Response:
I appreciate your suggestion for a deeper exploration of DLVO theory's applicability. I have now added further details regarding DLVO theory’s relevance to my PMMA colloidal system in Lines 202–206. Specifically, I clarified that the system involves deionized aqueous dispersions with negatively charged PMMA particles, which typically conform to DLVO theory. Additionally, I discussed the complex dependency of DLVO potential curves on particle size, zeta potential, and electrolyte concentration, emphasizing that flocculation does not occur without an electric field.
Comment 3: Visual Evidence of Particle Aggregation Under Electric Field
The mechanism behind particle flocculation induced by electric fields is discussed but not visually demonstrated.
Response:
Thank you for suggesting the inclusion of visual evidence. At this stage, I do not have direct visual evidence of particle flocculation under an electric field. However, indirect evidence of aggregation was documented in my previous study [9], as cited in Lines 397–398. Future research will prioritize obtaining direct visual evidence to further substantiate these findings.
Comment 4: Quantitative Analysis of Convective Patterns
Incorporate quantitative analysis (e.g., computational fluid dynamics simulations or tracer particles) to directly measure and validate convection patterns.
Response:
I acknowledge your suggestion to incorporate quantitative analysis through computational fluid dynamics (CFD) simulations or tracer particles. However, the main goal of this manuscript is to report a novel observed phenomenon rather than providing its quantitative characterization. Additionally, performing CFD simulations falls outside my current expertise. I agree that tracer particles represent a promising method for future quantitative convection analysis, and I explicitly note this direction as an important area for future research.
Thank you again for your insightful comments, which have significantly enhanced the clarity and quality of my manuscript. I hope my responses have adequately addressed your concerns.
Sincerely,
Hiroshi Kimura
Round 2
Reviewer 1 Report
Comments and Suggestions for Authors
I appreciate the effort of the author to clarify the points raised in the initial review stage; however, fundamental points are still present and then I have to reiterate my recommendation. Please see below:
- The motivation should be more solid. In my opinion, the author should state the originality of the work in respect to previous publications and/or reviews in the topic.does not
- The absence of rotation of the particles during the sedimentation should be supported.
- Additional perspectives were omitted
A proofreading is suggested
Author Response
Response to Reviewer 1
Dear Reviewer,
I sincerely appreciate your valuable comments and suggestions, which have significantly contributed to improving our manuscript. Below, we provide our responses to your comments and explain the corresponding revisions made to the manuscript.
- Comment: The motivation should be more solid. In my opinion, the author should state the originality of the work in respect to previous publications and/or reviews in the topic.does not
Response: There have been very few reports on the ERS effect to date, and most of them have been conducted by the author. The most relevant study is that of Mori et al. [24], who cited the author’s previous work on the ERS effect in their paper. Therefore, other related studies, mainly those conducted in confined spaces, have already been introduced in Lines 72–91. The primary difference between those studies and the present work is that this study specifically investigates "sedimenting particles" and, for the first time, examines the ERS effect under a DC electric field whose direction varies from horizontal to vertical. This point has been added to Lines 134–138 and Lines 458–461.
- Comment: The absence of rotation of the particles during the sedimentation should be supported.
Response: When I received your previous comment, I understood it as referring to the rotation of flocs. However, I realized that I had not considered the rotation of individual particles. I have now added a discussion on the relationship between the rotational diffusion motion of individual PMMA particles and their sedimentation velocity in Lines 38–41 and Lines 439–442.
- Comment: Additional perspectives were omitted
Response: I have added a discussion on the potential for controlling dispersion states through the combination of electric and magnetic fields in Lines 477–480.
Once again, I appreciate the constructive feedback and believe that the revisions have improved the manuscript. I look forward to any further comments you may have.
Best regards,
Hiroshi Kimura
Round 3
Reviewer 1 Report
Comments and Suggestions for Authors
I agree with the reviewed version of the manuscript. I can reccomend this work for publication in present form
Comments on the Quality of English LanguageA proofreading is suggested